# Effect of Nd Element of Mg-Nd Binary Alloy on the Corrosion Resistance in Sulfate-Reducing Bacteria Solution

**DOI:** 10.3390/ma15248788

**Published:** 2022-12-09

**Authors:** Zhenhua Chu, Zhixin Zhang, Yuanqing Zhou, Jingxiang Xu, Xingwei Zheng, Ming Sun, Fang Wang, Zheng Zhang, Qingsong Hu

**Affiliations:** 1Shanghai Engineering Research Center of Hadal Science and Technology, College of Engineering, Shanghai Ocean University, Shanghai 201306, China; 2College of Science, Donghua University, Shanghai 201620, China; 3College of Mechanical Engineering, University of Shanghai for Science and Technology, Shanghai 200093, China

**Keywords:** sulfate-reducing bacteria, Mg-Nd binary alloys, corrosion, biofilm, Mg_12_Nd

## Abstract

In this paper, the corrosion resistances of Mg-Nd binary alloys with various contents of the neodymium (Nd) element in sulfate-reducing bacteria (SRB) were studied. In the SRB medium, the results of weight loss experiments showed that the increase in the Mg_12_Nd phase in the alloy increased the galvanic corrosion and the corrosion rate. However, when the continuous network distribution of the second phase formed, the corrosion resistance of the alloy improved. The biofilm was formed by the adhesion of the SRB. Meanwhile, the protection from the corrosion improved due to the corrosion products, which prevent the penetration of corrosive ions. On the other hand, the products of biological metabolism accelerated the corrosion of the matrix.

## 1. Introduction

Magnesium (Mg) alloy is one kind of material with excellent performance strengths, such as light weight, a high specific strength, and good casting. It has been used widely in the automotive and aerospace industries. As the lightest metallic structural material, the application of magnesium alloys in marine structural engineering materials is promising. However, lower corrosion resistance limits the magnesium alloy application. It is well-known that Nd is a common rare earth element as an addition in Mg alloys that improves their strength [1,2]. At the same time, researchers have found that the Nd element also has a positive effect in improving the corrosion resistance of the alloy [3]. As such, the development of Mg alloys with the Nd element as the prominent additional element is very promising.

The marine environment is a complex environment in terms of corrosion. As far as the applications of magnesium alloys in the marine environment go, aside from the corrosion damage caused by the seawater medium, the microorganisms in the marine environment can also affect the corrosion behavior of magnesium alloys. Microbial corrosion (MIC) refers to the phenomenon of corrosion caused by the life activities of microorganisms, which is an important corrosion form that leads to the failure of marine engineering materials [4,5]. SRB is one kind of anaerobic bacterium that causes MIC, which has been studied widely [6,7]. However, several theories for the corrosion mechanism of SRB have been proposed, including cathodic depolarization (CDT) [8], concentration battery [9], extracellular electron transfer (EET) [10], etc. Due to the complex environment of microbial corrosion, the exact corrosion mechanism of magnesium alloy is still unclear.

In this paper, Mg-1Nd, Mg-2Nd, and Mg-3Nd binary alloys were chosen. The effect of the microstructure and components of the alloys on the corrosion process in a sulfate-reducing bacteria solution was studied. The corrosion properties were tested by electrochemical and immersion experiments. The results revealed the corrosion behavior of Mg-Nd binary alloys with different Nd contents. Further, the corrosion mechanism of Mg-Nd alloys caused by SRB is proposed.

## 2. Experimental Process

### 2.1. Material Preparation

Mg-1Nd, Mg-2Nd, and Mg-3Nd binary alloys were prepared in a resistance furnace. The whole metallurgical process was under the protection of SF_6_ plus CO_2_ mixture gas. Pure magnesium (99.9%) and Mg-30% Nd master alloys (99.5%) were used as preparation materials, and a JDMJ refining agent was used. The casting temperature of the alloy was 720~740 °C, and the casting was carried out in the metal mold with a mold temperature of 160 ± 10 °C.

### 2.2. Material Characterization

The composition of the cast alloys was tested with an inductively coupled plasma spectrometer (ICP), and the results are shown in Table 1. The microstructure of the alloy was observed by scanning electron microscopy (SEM) and imaged in backscattered electron mode. The phase composition of the alloy was determined by X-ray diffraction (XRD) at a speed of 5°/min.

### 2.3. Immersion Test

Three disk specimens were used for the immersion test, and the average value was taken. The disk size was ϕ 30 mm × 3 mm. The SRB solution was used as a corrosion medium, soaking for 7 days. The temperature was constant at 37 °C. After the experiment, the corrosion products were removed in a chromic acid (200 g/L CrO_3_ + 10 g/L AgNO_3_) solution and soaked for about 7 min [11]. The corrosion morphologies of the alloys were observed by SEM.

### 2.4. Electromechanical Corrosion Test

Electrochemical impedance spectroscopy (EIS) and potentiodynamic polarization curves were measured using a Gamry Ref 600+. A three-electrode system was adopted. The reference electrode was saturated calomel electrode, and the platinum sheet electrode was the counter electrode. The size of the tested sample was 10 mm × 10 mm × 3 mm. The other surfaces were sealed with epoxy resin, and the electrolyte solution was a SRB solution. An open circuit potential test was performed before the polarization curve and impedance spectroscopy measurements. The frequency range of the EIS was 0.1~100,000 Hz, and the amplitude was 5 mV. Polarization curves were measured in the corrosion potential (Ecorr vs. SCE) range of −0.3~0.3 V with a scan rate of 1 mV/s.

## 3. Results

### 3.1. Microstructure Characteristics

The microstructure of the Mg-Nd binary alloys was observed. It was a typical dendritic structure, which was composed of matrix with α-Mg phase (as shown in Figure 1, gray regions) and a second phase (as shown in Figure 1, white regions) [12]. The distribution of the second phase in the alloy changed with an increase in Nd content. For the Mg-1Nd alloy, the second phase, presenting a dotted distribution, was concentrated in the grain boundary. For Mg-2Nd alloy, the second phase was distributed in the grain boundary with a short rod-like and discontinuous shape. A continuous network structure of the second phase was formed in the Mg-3Nd alloy.

In order to further analyze the characteristics of the second phase of the alloy, X-ray diffraction (XRD) was adopted, and the diffraction patterns are shown in Figure 2. The results show that the diffraction peaks came from the α-Mg matrix and Mg_12_Nd. Therefore, the second phase of the alloy was determined as Mg_12_Nd [13]. It is worthy to note that with the increase in Nd content, the diffraction peak of the Mg_12_Nd phase gradually increased. Since the Nd content in the designed Mg-1Nd binary alloy was relatively small, and the maximum was only 3%, most of the peaks were from the α-Mg.

### 3.2. Immersion Test

The macroscopic morphologies removing the corrosion products of the Mg-Nd alloys after soaking in SRB solution for 7 days were observed, as shown in Figure 3. The results showed that pits appeared on the surfaces of all three samples. Many pits with different sizes and depths appeared on the surface of the Mg-2Nd alloy. This indicates that the local corrosion was the most serious.

The corrosion rates of the three alloys immersed for 7 days were calculated to investigate the effect of the Nd element on the corrosion resistance, and the results are shown in Figure 4. Comparing the three kinds of alloys, the average corrosion rate of the Mg-2Nd alloy was the largest, followed by the Mg-3Nd alloy, and the average corrosion rate of the Mg-1Nd alloy was the smallest. The rates were: 0.73 mg·cm^−2^·d^−1^, 1.13 mg·cm^−2^·d^−1^, and 0.81 mg·cm^−2^·d^−1^, respectively. This indicates that with the increase in Nd content, the corrosion rate of the alloy in the SRB medium first increased and then decreased. The immersion experiment showed that the binary alloy with 1% of the Nd element had the best corrosion resistance in the SRB medium, and the worst corrosion resistance was obtained for the alloy with 2% Nd.

The corrosion products of the Mg-2Nd alloy after immersion for 7 days were tested by XRD. The result is shown in Figure 5. It is shown that the corrosion products covered on the surface were Mg(OH)_2_. This comes from the corrosion reaction of the α-Mg matrix.

### 3.3. Potentiodynamic Polarization Analysis

The potentiodynamic polarization curves of the alloys in the SRB medium were tested, as shown in Figure 6. The corrosion potential (Ecorr) and the corrosion current density (icorr) of the alloy were obtained by Tafel fitting. The results are listed in Table 2. Generally, the alloy with a smaller corrosion current density and a higher corrosion potential showed better corrosion-resistance properties. Meanwhile, the cathode branches of the Tafel curves of the three alloys represent the hydrogen evolution, and the anode branches represent the dissolution of the magnesium matrix. Further, the anode branches of the Mg-1Nd and Mg-3Nd alloys show obvious current platforms [14]. In this platform, the current density increases slowly in a very narrow potential range and then increases rapidly with the increase in potential. However, a platform of current is observed in the anode part of Mg-2Nd. The existence of a current platform shows that the protective film formed on the Mg-1Nd and Mg-3Nd allows was better than that on the Mg-2Nd alloy.

### 3.4. Electrochemical Impendence Spectroscopy

The EIS of the Mg-Nd binary alloys immersed in the SRB medium for 1 day and 3 days was also investigated. According to the Nyquist plots, as shown in Figure 7a, it was found that with the increase in Nd content, the capacitive reactance arc radius of the Mg-Nd binary alloy first decreased and then increased. On the other hand, with the increase in soaking time, the radius increased. The arc radius of high frequency capacitive reactance is related to the charge transfer resistance and the formation of the corrosion product film. As such, the increase in the capacitive reactance arc radius shows that the impedance value of the corrosion product film increased. In other words, this means that the protective ability of the corrosion product film was enhanced.

The Bode impedance plots show two continuous semi-circle shapes for all the samples (as shown in Figure 7b). This means that there are two time constants. The time constant in the high-frequency range is related to the capacitance impedance of the biofilm. The other constant is inductive reactance, which is caused by the corrosive products of the alloy reaction process of the SRB.

According to the Bode plot, the EIS curves were fitted by the proposed equivalent circuit, as shown in Figure 8. The equivalent circuit is composed of the polarization resistance (Rp), the solution resistance (Rs), the biofilm resistance (Rf), the charge transfer resistance (Rct), the constant phase element in parallel (CPEf), the constant phase element in parallel (CPEdl), and the relevant inductance (L). The fitting results are summarized in Table 3. The Rf and Rct of the Mg-1Nd alloy were the largest, which shows that the charge transfer was the most difficult in the corrosion process and had the best protective effect of the alloy biofilm. This indicates that the alloy was the toughest to erode. With the increase in the immersion time, the Rct and the film resistance of the alloy increase, which indicates that the biofilm on the alloy surface protects the surface from corrosion.

### 3.5. Surface Morphology Analysis

The morphologies of the alloys immersed in the SRB medium for 3 days are shown in Figure 9. Many SRB bacteria became attached to the surfaces of the alloys, as shown in Figure 9d–f, which are the enlarged images marked in Figure 9a–c. Biofilms also formed on the surface. Comparing these three kinds of alloys, the number of bacteria in the Mg-2Nd alloy was the largest. In fact, the biofilm was composed of SRB bacteria, extracellular polymeric substances (EPS), and metabolites generated by SRB.

The corrosion morphologies of the Mg-Nd binary alloy soaked for 3 days and the corrosion products that were removed were observed, as shown in the Figure 10. Based on the results of XRD, the composition phases of the Mg-Nd binary alloy were the α-Mg phase and the Mg_12_Nd phase. Meanwhile, we noted that the Mg_12_Nd phase had higher potential. This is to say that it possesses better corrosion resistance than the α-Mg matrix phase. Therefore, it can be seen from Figure 10b that the α-Mg phase of the Mg-2Nd alloy was eroded completely, exposing the point-shaped and rod-shaped Mg_12_Nd phase. The corrosion areas expanded on the surface and connected with each other. Similar corrosion morphologies were obtained in the Mg-1Nd alloy and the Mg-2Nd alloy. However, for the Mg-3Nd alloy, the skeleton-like Mg_12_Nd phase was exposed, and the corrosion area was not interconnected like the Mg-2Nd alloy, which indicates that the skeleton-like Mg_12_Nd phase acted as a corrosion barrier to hinder the expansion of the corrosion area [15]. There are many small pits on the surface (as shown in Figure 10d), which were caused by the corrosion of HS^−^ [16].

According to the above results, the corrosion resistance of the Mg-1Nd and Mg-3Nd alloys in the SRB medium was better than that of the Mg-2Nd alloys. This is because the content of the second phase in the Mg-1Nd alloy was less than that of the Mg-2^Nd^ alloy, and the internal galvanic corrosion was weaker than that of the Mg-2Nd alloy. The second phase with a network distribution in the Mg-3Nd alloy was not conducive to the expansion of corrosion, which improved the corrosion resistance.

## 4. Discussion

According to the above results, the corrosion process and corrosion mechanisms of Mg-1Nd, Mg-2Nd, and Mg-3Nd binary alloys in SRB solution are proposed. The corrosion demonstration is shown in Figure 11. At the early stage of immersion, the SRB bacteria are first adsorbed to the surface of alloy. At this stage, due to the second phase of the Mg_12_Nd with higher electrode potential than the α-Mg matrix, the reaction Mg → Mg^2+^ + 2e^−^ is promoted. This is a cathodic reaction for the SRB’s corrosion reaction. Therefore, with the increase in the Mg_12_Nd phase, the adsorption of the SRB is enhanced.

Then, with the increase in immersion time, biofilms are formed by EPS, and immobilized bacteria are formed on the alloy surface. At the same time, the alloy’s surface is covered with biofilms, leading to an anoxic environment. According to the cathodic depolarization theory (CDT), in an oxygen-deficient environment, the [H] is generated by a reduction in the H^+^ covering the surface of the alloy. The cathodic reaction is carried out by the SRB to reduce the sulfate. Therefore, the SRB promotes the cathodic reaction to accelerate the corrosion of the alloy. On the other hand, the HS^−^ and acidic organics, which are generated in the process of the SRB metabolism, also promote the anodic dissolution of the alloy. This results in the formation of severe localized corrosion.

According to the cathodic depolarization theory, the reaction process of the Mg-Nd binary alloy in the SRB medium may be as follows:(1)$$\mathrm{cathodic}\;\mathrm{reaction}:\;\mathrm{Mg}\;\to \;{\mathrm{Mg}}^{2+}+2e$$
(2)$$\mathrm{anode}\;\mathrm{reaction}:\;H_{2}O\to H^{+}+{\mathrm{OH}}^{-}$$
(3)$$\mathrm{cathodic}\;\mathrm{depolarization}:\;H^{+}+e\to H$$
(4)$${\mathrm{SO}}_{4}^{2-}+8H+H^{+}\to {\mathrm{HS}}^{-}+4H_{2}O$$
(5)$$H^{+}+{\mathrm{HS}}^{-}\to H_{2}S$$
(6)$$\sec\mathrm{ondary}\;\mathrm{reaction}:\;{\mathrm{Mg}}^{2+}+2{\mathrm{OH}}^{-}\to \mathrm{Mg}{\left(\mathrm{OH}\right)}_{2}$$

It is worthwhile to note that the corrosion process is related to the distribution of the second phase of Mg_12_Nd. Although the metabolic reaction of the sulfate-reducing bacteria is accelerated due to the formation of Mg_12_Nd. When the distribution of the network is formed, the diffusion of the corrosion product is hindered. The penetration of corrosive ions is blocked. Therefore, the corrosion rate slows.

## 5. Conclusions

In this study, the corrosion resistance of Mg-1Nd, Mg-2Nd, and Mg-3Nd binary alloys were studied. Some conclusions are summarized as follows:(1)With the increase in the Nd element, the average corrosion rate of the Mg-Nd binary alloy in the SRB medium increases first and then decreases. This is related to the distribution of the second phase. The corrosion process is accelerated only when the distribution of the second phase is discontinuous. However, the continuous network distribution of the second phase can improve the corrosion resistance.(2)The corrosion characteristics of the Mg-Nd binary alloys in the SRB medium pit the corrosion caused by galvanic corrosion. A biofilm can be formed on the surface, and the Nd element affects the protection of the biofilm.(3)There are two functions for the biofilm formed on the surface. One is to slow down the process of corrosion by blocking the penetration of erosive ions. The other function is the biological metabolism of the HS^−^ ions and organic acids generated by SRB, which corrode the alloy matrix.

## Figures and Tables

**Figure 1 materials-15-08788-f001:**
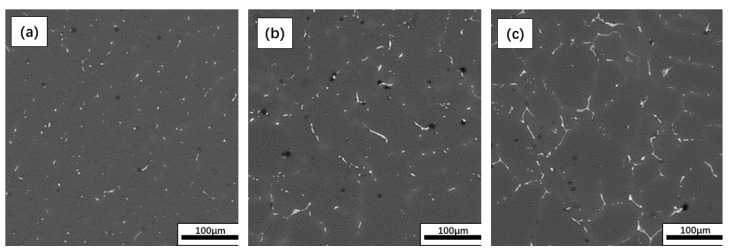
Backscatter plot of F-state Mg-Nd alloys: (**a**) Mg-1Nd, (**b**) Mg-2Nd, (**c**) Mg-3Nd.

**Figure 2 materials-15-08788-f002:**
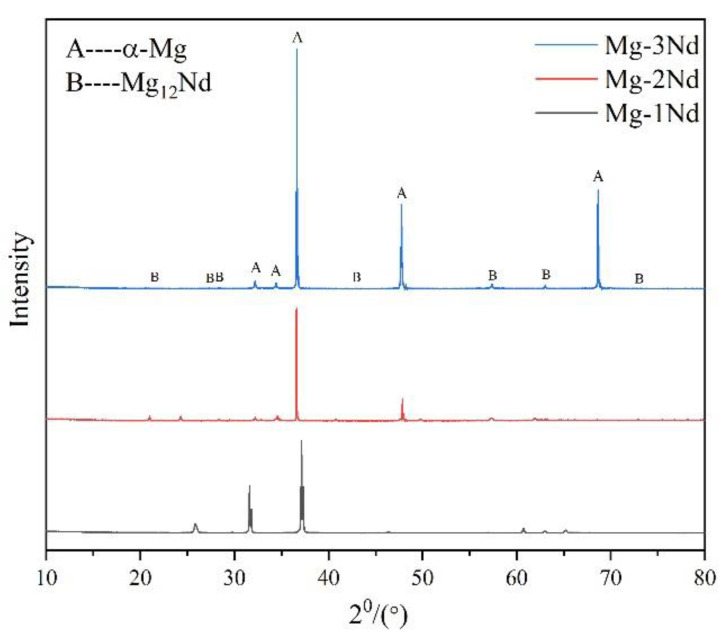
XRD patterns of three Mg–-Nd alloys.

**Figure 3 materials-15-08788-f003:**
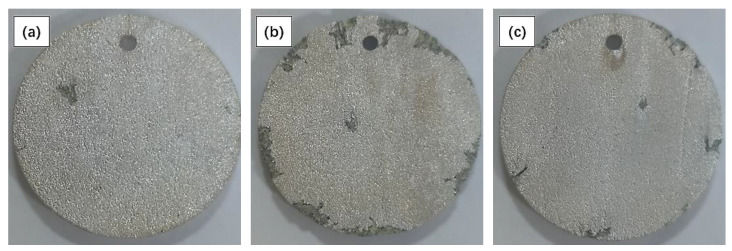
Macroscopic topographies of Mg-Nd alloy after soaking for 7 days in F-state Mg-Nd alloy: (**a**) Mg-1Nd, (**b**) Mg-2Nd, (**c**) Mg-3Nd.

**Figure 4 materials-15-08788-f004:**
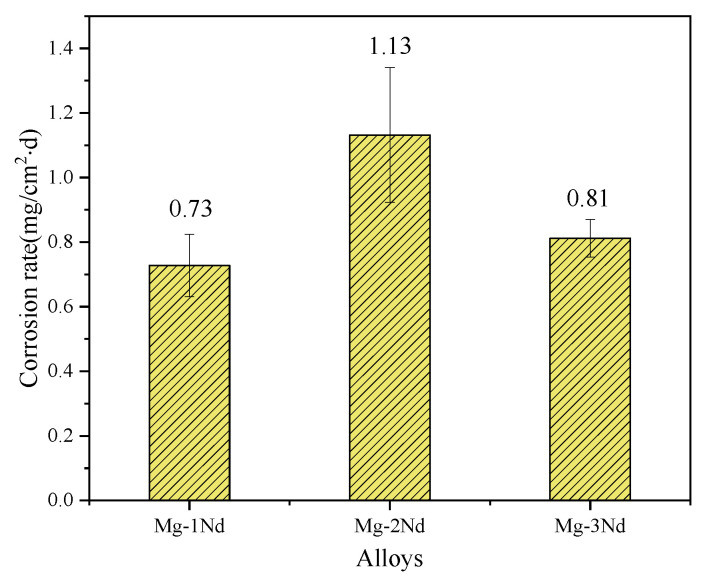
Corrosion rate of F-state Mg-Nd alloy after immersion for 7 days.

**Figure 5 materials-15-08788-f005:**
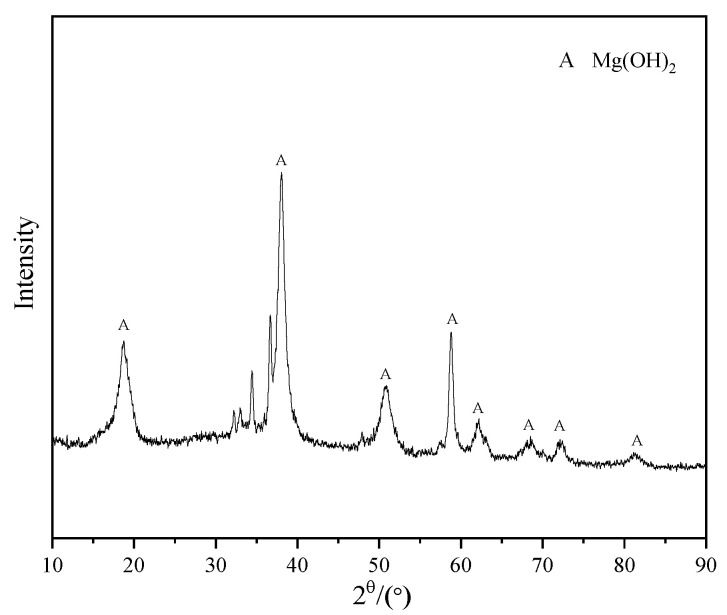
XRD patterns of corrosion products of F-state Mg-2Nd alloy.

**Figure 6 materials-15-08788-f006:**
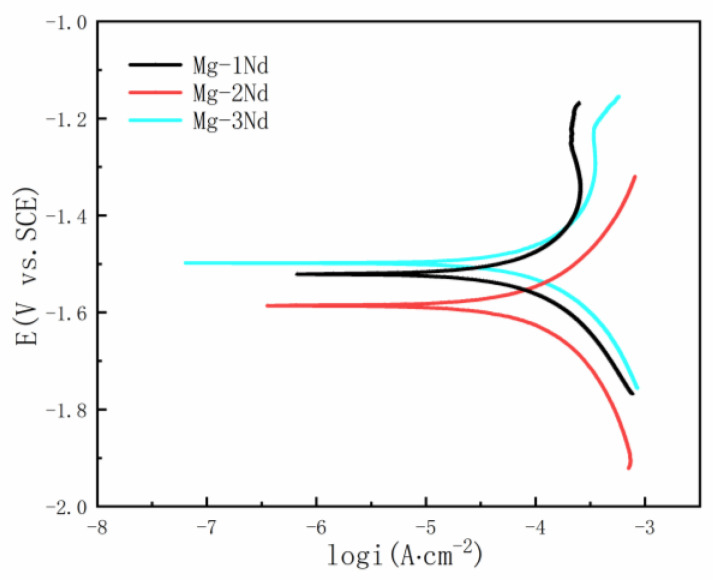
Potentiodynamic polarization curves of F-state Mg-Nd alloy in SRB.

**Figure 7 materials-15-08788-f007:**
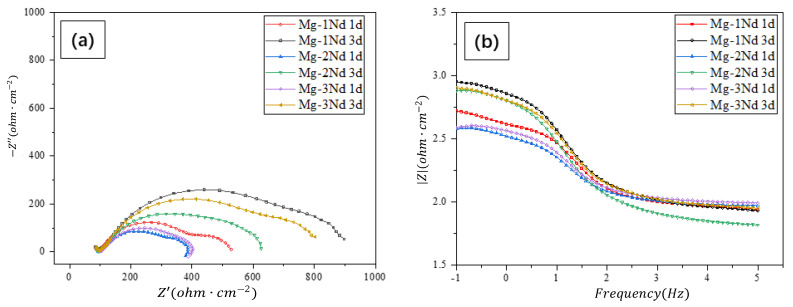
(**a**) Nyquist diagram of Mg-Nd alloy in F state; (**b**) relationship between impedance and frequency in Bode diagram.

**Figure 8 materials-15-08788-f008:**
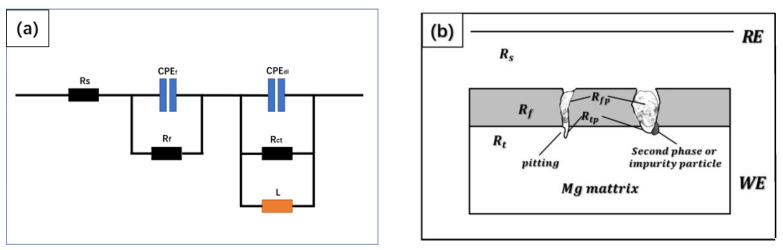
(**a**) Equivalent circuit diagram; (**b**) physical model of equivalent circuit.

**Figure 9 materials-15-08788-f009:**
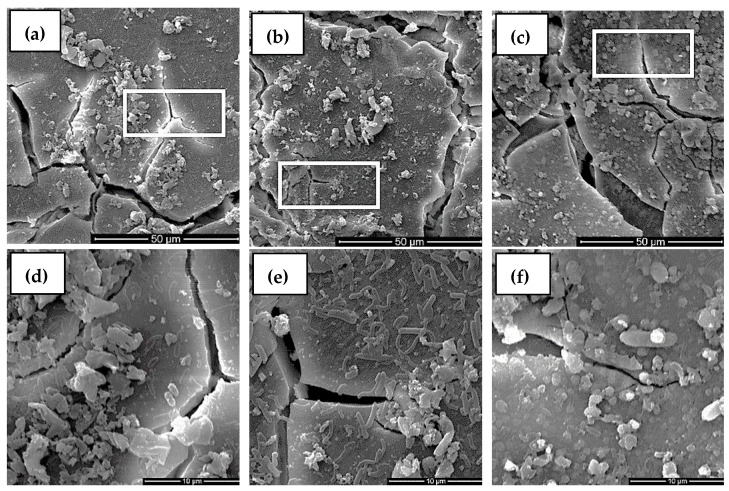
Morphologies of alloys immersed in SRB for 3 days: (**a**) Mg-1Nd alloy, (**b**) Mg-2Nd alloy, (**c**) Mg-3Nd alloy; local enlarged view of: (**d**) Mg-1Nd alloy, (**e**) Mg-partial, (**f**) Mg-3Nd alloy.

**Figure 10 materials-15-08788-f010:**
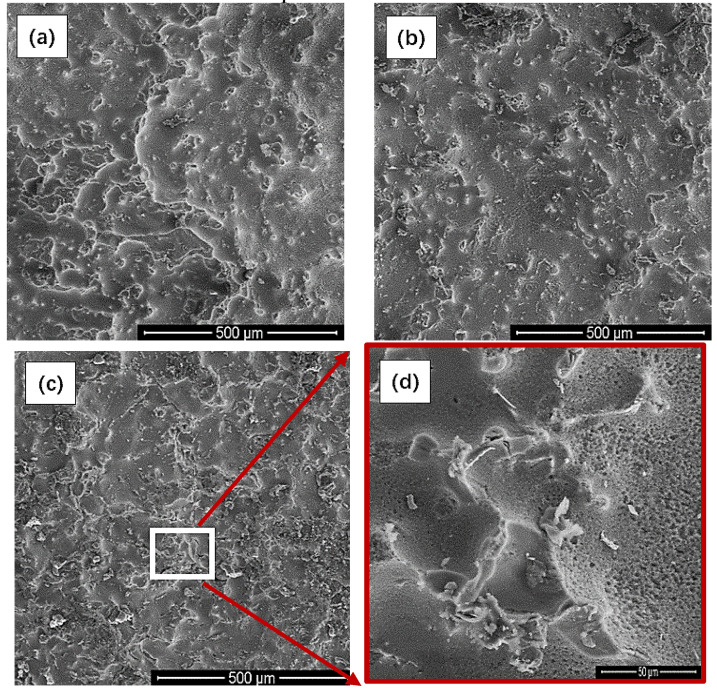
Morphologies of the alloys immersed for 3 days to remove corrosion products: (**a**) Mg-1Nd alloy, (**b**) Mg-2Nd alloy, (**c**) Mg-3Nd alloy; (**d**) local magnification of Mg-3Nd alloy picture.

**Figure 11 materials-15-08788-f011:**
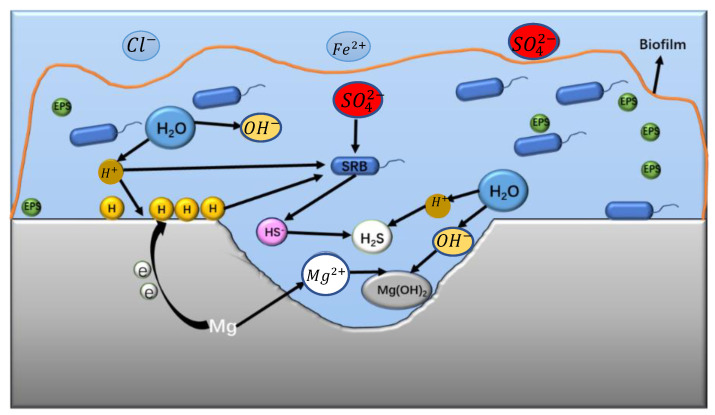
Corrosion mechanism diagram of Mg-Nd alloy in SRB.

**Table 1 materials-15-08788-t001:** Measured composition of the alloy.

Alloy Code	Chemical Composition, wt.%		
	Nd	Fe	Mg
Mg-1Nd	1.172	0.012	margin
Mg-2Nd	1.997	0.031	margin
Mg-3Nd	2.968	0.025	margin

**Table 2 materials-15-08788-t002:** Polarization curve fitting parameters for F-state Mg-Nd alloys.

Alloys	Ecorr (V)	Icorr (μA·cm^−2^)	Corrosion Rate (mmpy)
Mg-1Nd	−1.521	107	1.242
Mg-2Nd	−1.586	175.9	2.041
Mg-3Nd	−1.498	130.4	1.514

**Table 3 materials-15-08788-t003:** EIS fitting results of F-state Mg-Nd alloys.

Time (day)	Rs (Ω/cm^2^)	Rf (Ω/cm^2^)	CPEf (mF/cm^2^)	nf	Rct (Ω/cm^2^)	CPEdl (mF/cm^2^)	ndl	L (H/cm^−2^)
Mg-1Nd								
1	54.61	448.5	190.2 × 10^−6^	631.1 × 10^−3^	35.43	30.56 × 10^−9^	946.6 × 10^−3^	38.46 × 10^−3^
3	14.06	900.4	123.5 × 10^−6^	671.3 × 10^−3^	76.38	31.69 × 10^−9^	880.0 × 10^−3^	245.5 × 10^−3^
Mg-2Nd								
1	73.32	319.4	287.0 × 10^−6^	638.9 × 10^−3^	21.60	439.5 × 10^−9^	725.8 × 10^−3^	38.20 × 10^−3^
3	52.73	600.7	230.3 × 10^−6^	638.3 × 10^−3^	41.73	4.454 × 10^−6^	468.6 × 10^−3^	114.4 × 10^−3^
Mg-3Nd								
1	70.64	334.6	178.0 × 10^−6^	708.9 × 10^−3^	33.10	93.32 × 10^−6^	325.9 × 10^−3^	99.10 × 10^−3^
3	32.23	784.7	146.6 × 10^−6^	655.6 × 10^−3^	60.58	17.33 × 10^−9^	954.6 × 10^−3^	153.7 × 10^−3^

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
