# Peer review of "Effect of Nd Element of Mg-Nd Binary Alloy on the Corrosion Resistance in Sulfate-Reducing Bacteria Solution"

_materials, 2022, doi:10.3390/ma15248788_

Round 1
Reviewer 1 Report
The manuscript studied the effect of the Nd element on the formation of the biofilm of Mg-Nd binary alloy in a sulfate-reducing bacteria solution. The theme of the paper is sound and exciting. The paper is well-organized and straightforward. I recommend this paper for publication in materials after minor revision of the following points.
1) A specialist in the English language must review the manuscript because it contains many linguistic errors that cannot be listed here.
2) The chemical equations must be written by the equation editor in the MSWord.
3) The authors must clarify why the rate of Mg-2Nd alloy is the largest
Author Response
1) A specialist in the English language must review the manuscript because it contains many linguistic errors that cannot be listed here.
As suggested by the reviewer, the English has been modified carefully one by one in the revised manuscript.
2) The chemical equations must be written by the equation editor in the MSWord.
As suggested by the reviewer, the equations have been re-written.
3) The authors must clarify why the rate of Mg-2Nd alloy is the largest
As pointed by the reviwer, the corrosion mechnism is important. And we have added the explanation in the disscution in the revised manuscript.
In short, the Mg12Nd phase is formed due to the addition of Nd element. Due to the second phase Mg12Nd with higher electrode potential than α-Mg matrix, the reaction Mg → Mg2+ + 2e- is promoted. It is cathodic reaction for SRB corrosion reaction. Therefore, with the increase of Mg12Nd phase, the adsorption of SRB is enhanced. On the other hand, it is worthy to note that the corrosion process is related to the distribution of the second phase Mg12Nd. Although the metabolic reaction of sulfate reducing bacteria is accelerated due to the formation of Mg12Nd. When the distribution of network is formed, the diffusion of corrosion product is hindered. The penetration of corrosive ions is blocked. Therefore the corrosion rate is slowed. So the Mg12Nd phase is discontinuous distribution in Mg-2Nd alloy with the highest corrosion rate.
Reviewer 2 Report
English should be improved. Please provide EDS data justifying formation of corrosion products after 7 days soaking.
Author Response
1.English should be improved.
As suggested by the reviewer, the English has been modified carefully one by one in the revised manuscript.
2.Please provide EDS data justifying formation of corrosion products after 7 days soaking.
It is a good suggestion. And we have tried to get a perfect EDS data. The result of EDS show that the corrosion product is composed of Magnesium and oxygen elements. However, according to our judgment, the hydrogen element should be contained in the corrosion product. So we provide the XRD result to explain the composition of the corrosion product in the revised manuscript.
Reviewer 3 Report
1. Authors try to develop new Mg-Nd binary alloys. But at present a high amount of alloys with Nd and other components are known. The necessity to develop such an alloy is doubtful.
2. Please provide the purity of raw materials in 2.1 section.
3. 2.4 section: Must be mV/s instead of mv/s.
4. 3.1 Section: The color of magnesium matrix is not black; it is gray.
5. Page 3. Please use subscript symbols for 12 in Mg12Nd phase.
6. Fig. 5: Please change the color of polarization curve for Mg-3Nd alloy.
7. In section 3.6 the mechanical properties of alloy is provided, but no information about the mechanical test is provided in the materials and methods section. What is the samples size, and used equipment? How did the mechanical samples corrode for 3 days?
8. The discussion is not about the results obtained in the article and must be rewritten.
Author Response
1. Authors try to develop new Mg-Nd binary alloys. But at present a high amount of alloys with Nd and other components are known. The necessity to develop such an alloy is doubtful.
As pointed by the reviewer, the Mg alloys with other components are known. However, the corrosion mechanism of Mg alloy, especially the corrosion mechanism of sulfate reducing bacteria, which is one kind of anaerobic bacteria in deep sea environment, hasn’t been clarified. As far as the development of Mg alloys used for deep marine engineering, it should be clarified the corrosion resistance. According to the present study, we chose Nd element which is popular as an addition element in Mg alloys to study the effect on the corrosion mechanism in SRB solution, firstly. And then, some other elements are going to be investigated in the next study.
2. Please provide the purity of raw materials in 2.1 section.
They have been added in the revised manuscript.
3. 2.4 section: Must be mV/s instead of mv/s.
It has been corrected in the revised manuscript.
4. 3.1 Section: The color of magnesium matrix is not black; it is gray.
As pointed by the reviewer it has been corrected in the revised manuscript.
5. Page 3. Please use subscript symbols for 12 in Mg12Nd phase.
As pointed by the reviewer, it has been modified in the revised manuscript.
6. Fig. 5: Please change the color of polarization curve for Mg-3Nd alloy.
As suggested the reviewer, the Fig. 5 has been re-drawn. The curve of Mg-3Nd alloy has been bolded.
7. In section 3.6 the mechanical properties of alloy is provided, but no information about the mechanical test is provided in the materials and methods section. What is the samples size, and used equipment? How did the mechanical samples corrode for 3 days?
Yes. As pointed by the reviewer, the mechanical experiment process should be added. However, we re-written the manuscript. And we think that the mechanical test isn't necessary to explain the corrosion mechanism. It is better to be published in the paper to discuss the mechanical results. So we delect the results.
8. The discussion is not about the results obtained in the article and must be rewritten.
As suggested by the reviewer, we rewritten the discussion part in the revised manuscript.
Round 2
Reviewer 3 Report
The reviewers comments are addressed well.